# Clinical Impact of Measurable Residual Disease in Acute Myeloid Leukemia

**DOI:** 10.3390/cancers14153634

**Published:** 2022-07-26

**Authors:** Tali Azenkot, Brian A. Jonas

**Affiliations:** 1Department of Internal Medicine, University of California Davis School of Medicine, Sacramento, CA 95817, USA; tazenkot@ucdavis.edu; 2Division of Cellular Therapy, Bone Marrow Transplant, and Malignant Hematology, Department of Internal Medicine, University of California Davis School of Medicine, Sacramento, CA 95817, USA

**Keywords:** AML, acute myeloid leukemia, MRD, measurable residual disease, hematopoietic stem cell transplant

## Abstract

**Simple Summary:**

Advances in immunophenotyping and molecular techniques have allowed for the development of more sensitive diagnostic tests in acute leukemia. These techniques can identify low levels of leukemic cells (quantified as 10^−4^ to 10^−6^ ratio to white blood cells) in patient samples. The presence of such low levels of leukemic cells, termed “measurable/minimal residual disease” (MRD), has been shown to be a marker of disease burden and patient outcomes. In acute lymphoblastic leukemia, new agents are highly effective at eliminating MRD for patients whose leukemia progressed despite first line therapies. By comparison, the role of MRD in acute myeloid leukemia is less clear. This commentary reviews select data and remaining questions about the clinical application of MRD to the treatment of patients with acute myeloid leukemia.

**Abstract:**

Measurable residual disease (MRD) has emerged as a primary marker of risk severity and prognosis in acute myeloid leukemia (AML). There is, however, ongoing debate about MRD-based surveillance and treatment. A literature review was performed using the PubMed database with the keywords MRD or residual disease in recently published journals. Identified articles describe the prognostic value of pre-transplant MRD and suggest optimal timing and techniques to quantify MRD. Several studies address the implications of MRD on treatment selection and hematopoietic stem cell transplant, including patient candidacy, conditioning regimen, and transplant type. More prospective, randomized studies are needed to guide the application of MRD in the treatment of AML, particularly in transplant.

## 1. Introduction

Acute myeloid leukemia (AML) is a hematologic malignancy characterized by the clonal expansion of undifferentiated myeloid precursors, resulting in impaired hematopoiesis and bone marrow failure. Although 70% of patients with AML attain morphologic complete remission (CR) with induction chemotherapy, approximately 50% of these patients experience relapse [1]. Overall survival in AML remains a disappointing 24% at 5 years [2].

The classification of AML has historically been based on cell morphology. The advent of new flow cytometry and molecular techniques have allowed for the identification of phenotypic and genetic markers present in leukemic cells. In 2017, an expert panel of the European Leukemia Net (ELN) proposed risk stratification criteria for AML based on genetic mutations at diagnosis [3]. The following year, ELN recommended a standardization of emerging techniques that test for measurable residual disease (MRD): the presence of low levels of leukemic cells after treatment, usually at 10^−4^ to 10^−6^ ratio to white blood cells [4]. Since then, MRD has become a key prognostic biomarker and tool to anticipate relapse in AML [5,6,7,8,9]. In response to advances in MRD detection, quantification, and application in patient care, ELN and the National Comprehensive Cancer Network (NCCN) updated recommendations on the use of MRD in AML in 2021 [10,11].

Despite these guidelines, many questions remain about the application of MRD in treatment of both fit patients, who undergo high-intensity chemotherapy (i.e., the cytarabine and daunorubicin-based 7 + 3 or CPX-351), and “unfit” patients, who are treated with low-intensity chemotherapy (i.e., hypomethylating agents or low-dose cytarabine). For unfit patients, the toxicity of high-intensity treatments outweighs the benefits due to age, comorbidities, or functional status [3,9]. How and when should patients be tested for MRD? Can we use MRD as a predictor of impending relapse, such that we should change treatment in response to its identification? How should the presence of MRD impact treatment decisions? For example, the bispecific T-cell engager blinatumomab was the first widely applied therapeutic agent to target MRD and improve survival in relapsed/refractory acute lymphoblastic leukemia [12]. Is there a parallel intervention for patients with AML? Herein, we comment on evidence and remaining questions about the treatment of patients with AML in relation to MRD status.

## 2. Methods

Articles included here were identified by searches on the PubMed database. Search terms included “MRD OR residual disease”, “AML OR myeloid leukemia”, and “transplant*”. Articles were identified from January 2016 to April 2022 and restricted to the English language. Expert opinion was included based on clinical experience in treating patients with AML at the University of California, Davis Comprehensive Cancer Center.

## 3. MRD Testing

### 3.1. How Should We Test Patients for MRD?

In line with ELN, NCCN, and American Society of Clinical Oncology guidelines, most centers use a combination of methods—including bone marrow morphology, cytogenetics, fluorescence in situ hybridization (FISH), immunophenotyping, and molecular studies—to optimize AML diagnostics and MRD technique sensitivity, specificity, and applicability [10,13,14,15]. Among these techniques, the most commonly used MRD detection tool is immunophenotyping with multiparameter flow cytometry (MFC) [1,16,17,18,19,20,21,22]. ELN guidelines provide guidance regarding flow cytometry settings, the recommended panel of cell population identifiers, and gating strategy [22]. The two primary methods of MFC interpretation are known as leukemia-associated immunophenotype (LAIP) and different from normal (DfN) [23]. ELN recommends the continued integration of LAIP/DfN populations in MRD detection, as well as calculating the lower limits of detection and quantification for each assay [10]. The recommended threshold for test positivity is ≥0.1% [10].

Polymerase chain reaction (PCR)-based methods of MRD detection include quantitative PCR (qPCR) and the emerging technique of digital droplet PCR (ddPCR). qPCR has become a standard technique for MRD detection. When compared to qPCR, ddPCR is more timely, sensitive, and reproducible because it does not rely on a standardization curve or reference materials [5,24,25]. ddPCR does, however, require specific assays for each gene aberration, and validation is still needed for its use with complimentary DNA [10].

As known markers with available PCR assays account for only 40–60% of AML cases, further identification and standardization of these tests is ongoing [10]. Assays have been developed for three phenotypes of particular clinical significance: *NPM1*, Core Bind Factor (CBF)-AML (referring to the fusions gene *RUNX1::RUNX1T1* and *CBFB::MYH11*) and Acute Promyelocytic Leukemia (APL) (referring to the fusion gene *PML::RARA)* [10]. If detected, PCR monitoring of these mutations is preferred over MFC [10]. PCR can also be applied to the fusion genes *BCR::ABL1*, *KMT2::MLLT3*, and *DEK::NUP214* [26]. In addition, Wilms tumor-1 (*WT1*), found in more than 80% of AML cases, may be a marker of MRD when identified as overexpressed by PCR [27].

Next generation sequencing (NGS)—including whole genome sequencing, whole exome sequencing, and targeted-gene sequencing—is another primary modality of molecular MRD detection [15,28,29,30,31]. MRD-targeted NGS panels are recommended to include signaling pathway genes (i.e., *FLT3-ITD*, *FLT3-TKD*, *KIT*, *RAS*) and any molecular marker targeted in treatment (i.e., *FLT3* or *IDH1/IDH2*) [10]. The ELN’s proposed detection threshold for all molecular studies is 10^−3^ or lower [10].

Costs to patients and our healthcare systems must be considered with the use of multiple MRD detection techniques. To lessen invasive patient testing, several studies advocate for the use of peripheral blood rather than bone marrow samples [32,33,34]. In one retrospective cohort of 209 MRD-positive patients, peripheral blood MFC captured 83% of patients with MRD on bone marrow aspirates with a specificity of 95% [34]; a higher sampling frequency may mitigate the lower sensitivity in peripheral blood compared to bone marrow. As there is likely additive value in combining MFC and molecular studies [15], highly sensitive, combined techniques are being developed [35]. Expense and resource requirements remain significant barriers to the use of NGS among MRD quantification techniques [7,36].

### 3.2. At What Time Points Should We Test for MRD?

Practices vary regarding the frequency of MRD monitoring. ELN consensus recommends MRD monitoring at a minimum of: at diagnosis, after two cycles of standard induction or consolidation chemotherapy, and at the end of treatment [4]. This recommendation acknowledges that molecular MRD markers may evolve over time; the kinetics of specific mutations over the course of diagnosis, treatment(s), and relapse continue to be explored as MRD detection sensitivity improves [6,37]. For well-characterized AML phenotypes (including *NPM1-mutant*, CBF, and APL), follow-up testing is also recommended by bone marrow sampling every 3 months or peripheral blood every 4–6 weeks in the first 24 months after treatment completion [10]. At our institution, for unfit patients treated with low-intensity therapy, we recommend monitoring MRD, MFC, and qPCR of previously detected molecular markers (i.e., *NPM1*) in bone marrow every three treatment cycles until MRD response, followed by every 1–3 months in the peripheral blood [38]. As MRD is predictive of relapse, particularly in well-defined MRD markers such as fusion genes of CBF AML, some recommend even more frequent monitoring to allow for prompt intervention if MRD is identified [39,40,41,42]. Figure 1 proposes a framework for MRD monitoring in patients with AML treated with either high- or low-intensity therapy.

### 3.3. How Should We Account for Mutations Associated with Clonal Hematopoiesis?

Clonal hematopoiesis of indeterminate potential (CHIP) refers to the age-related recurrence of genetic mutations in healthy persons without hematologic disease [15,43]. Mutations commonly associated with CHIP may be an ancestral clone that gave rise to AML or a somatic clone in non-preleukemic genes [10,15]. In *NPM1*-mutated AML, the persistence of DTA (*DNMT3A*, *TET2*, and *ASXL1*) mutations have not been shown to correlate with age, the intensity of induction therapy, or relapse-free survival, and are therefore understood as likely distinct from MRD markers [15,44]. Studies on the clinical implication of non-DTA CHIP mutations and germline mutations are ongoing [45,46]. For example, recent studies have demonstrated the significance of mutation characteristics, such as variant allele frequency, location, co-mutations, and downstream involvement in gene processing, of genes *SRSF2* and *IDH2 R140Q* [47,48]. CHIP introduces more nuance in MRD detection, including both endogenous CHIP and donor-derived CHIP in patients who pursue allogeneic hematopoietic stem cell transplant (allo-HSCT) [43].

## 4. MRD in Initial Therapy

### 4.1. What Is the Significance of MRD-Positive Status at First Complete Remission?

MRD is a strong prognostic factor for poor outcomes in AML. The AML17 trial was a foundational study to support the prognostic impact of MRD in fit patients with *NPM1*-mutated AML. In this study, patients in MRD-positive complete remission (CR) had a significantly higher risk of relapse (82% vs. 30%) and lower rate of survival at 3 years (24% vs. 75%) compared to their MRD-negative counterparts [49]. In multivariate analysis, MRD status was the only significant prognostic marker for relapse and death. A subsequent meta-analysis using data from 11,151 patients in 81 studies demonstrated improved 5-year disease-free and overall survival in patients without versus those with MRD (respectively 64% vs. 25%, 68% vs. 34%); these results were significant across all age groups, AML subtypes, time of MRD assessment, and MRD assessment methods except for cytogenetics and FISH [50].

Increasing evidence supports the prognostic role of MRD in unfit patients treated with low- or semi-intensive chemotherapy. For example, in the PETHEMA-FLUGAZA trial, among patients treated with either semi-intensive chemotherapy (fludarabine and cytarabine) or 5-azacitidine, patients in MRD-positive CR had significantly worse cumulative incidence of relapse (hazard ratio, HR 2.95) and relapse-free survival (HR 3.45) when compared to those who achieved MRD negativity [51]. In a smaller cohort of patients treated with decitabine and venetoclax, patients who achieved MRD negativity had improved relapse-free survival (HR 0.31) and overall survival (HR 0.23) compared to those who were MRD-positive at 2 months after treatment initiation [52].

Though we presume the sooner the better, the prognostic significance of MRD conversion at specific timepoints during low-intensity chemotherapy treatment remains unknown. For example, in a retrospective analysis of the VIALE-A trial, patients who achieved MRD-negative CR after one cycle of venetoclax and azacitidine had a longer duration of response, event-free survival, and overall survival compared to patients who had detectable MRD at this timepoint; MRD-positivity was predictive of overall survival (HR 0.285) [38]. In addition to the 25% of patients who achieved MRD-negative status at cycle 1, an additional 54% of patients achieved MRD response after cycle 7 and had comparable outcomes [38]. In comparison, the QUAZAR AML-001 trial showed improved overall and relapse-free survival with oral azacitidine versus placebo in both MRD-positive (HR 0.69, 0.58) and MRD-negative (HR 0.81, 0.71) patients [53]. In patients with persistent MRD, one fourth converted to MRD-negative status after 6 months of azacitidine therapy [53]. More studies are needed to determine the optimal timing of MRD assessment and, after MRD conversion, whether/when treatment cessation or transition to monotherapy is appropriate in unfit patients.

The role of MRD in patients who receive treatment for relapsed/refractory (r/r) AML is less clear. One retrospective study in over 100 patients with r/r AML showed that patients who achieved MRD-negative CR after salvage therapy had longer median time to relapse and relapse-free survival than their counterparts in MRD-positive CR (10.6 vs. 4.7 months and 7.3 vs. 4.6 months, respectively) [54]. This difference was in part attributed to increased allo-HSCT in patients in MRD-negative CR.

### 4.2. What Treatment(s) Can Eradicate MRD?

Given the prognostic significance of MRD-positivity at first CR, MRD-targeted therapy may be indicated to improve patient outcomes independent of plans for transplant, using both known modalities and novel agents. Ongoing investigations explore the optimal timing and duration of commonly used modalities, such as chemotherapy [55] and HSCT. Though reviewed by Ball et al. [56], notable novel agents include the hypomethylating agents azacitidine and decitabine with or without venetoclax [38,57,58]; oncogenic driver-mutation-targeted treatments (i.e., the *IDH1* inhibitors ivosidenib and enasidenib, as well as the *FLT3* inhibitors midostaurin, gilteritinib, quizartinib, and sorafenib); and immunotherapies, such as gemtuzumab ozogamicin [59]. Figure 2 highlights the ongoing debate about the application of these MRD-targeted treatments to six clinical scenarios in AML treatment.

### 4.3. Can We Use MRD-Directed Treatment to Prevent Relapse after Initial Chemotherapy?

MRD relapse is the recurrence of prior or new markers of MRD by MFC or molecular techniques. Identification of MRD may allow for early interventions to prevent hematologic relapse. Few studies have demonstrated the impact of preemptive therapy in patients with persistent or relapsed MRD at the completion of initial chemotherapy [60].

For example, in the RELAZA-2 trial, patients with advanced myelodysplastic syndrome (MDS) or AML, half of whom had not undergone allo-HSCT, were treated with azacitidine if identified to be MRD-positive on surveillance testing in the 24 months after achieving CR [61]. At 12 months, overall and relapse-free survival were 75% and 46% in MRD-positive patients treated with azacitidine, versus 91% and 88% in MRD-negative patients, respectively. The authors conclude that pre-emptive therapy with azacitidine can prevent or at least delay hematologic relapse in MRD-positive patients.

In another prospective analysis of patients with *NPM1*-mutated intermediate risk AML, Bataller et al. identified favorable outcomes with pre-emptive treatment with high-dose cytarabine-based chemotherapy or azacitidine (with or without subsequent allo-HSCT) for patients with molecular failure, quantified by the persistence of high *NPM1* mutant allele ratio or other MRD reappearance [62]. They found that two-year overall survival was far superior (>80% vs. 40%) for patients who received this preemptive treatment versus those treated at hematologic relapse. In both studies, lead time bias may inaccurately portray prolonged survival in patients with MRD over molecular relapse.

## 5. MRD in Transplant and Beyond

### 5.1. Prognostic Value of MRD Prior to Transplant

The prognostic role of MRD prior to allo-HSCT was established over a decade ago [63]. Since then, several studies have since compared outcomes of patients transplanted with active disease and those transplanted in morphologic CR, as stratified by MRD status. A retrospective analysis of 359 adults with AML by Araki et al. demonstrated similar 3-year overall survival and relapse rates in patients transplanted with active disease versus MRD-positive CR (23% vs. 26%, 65% vs. 67%); compared to these cohorts, outcomes were improved in patients in MRD-negative CR (3 year overall survival 73%, relapse rate 22%) [17]. A similar study was performed by Jentzsch et al., wherein 392 patients in either MRD-negative CR, MRD-positive CR, or with active disease had progressively worse event-free survival after allo-HSCT [64].

The impact of MRD status on patients who undergo allo-HSCT in CR1 versus CR2 has also been studied. In one retrospective cohort of 253 patients, Walter et al. identified worse 3-year overall survival in CR1 and CR2 among MRD-positive (32%, 44%) versus MRD-negative (73%, 73%) patients [18]. Similarly, 3-year relapse rates were worse in both CR1 and CR2 among MRD-positive (58%, 68%) versus MRD-negative (21%, 19%) patients. When combining CR1 and CR2 groups, the risk of death and relapse were 2.61 and 4.90 times higher for patients in MRD-positive versus MRD-negative CR at transplant [18]. In a larger cohort of 1042 patients with AML in CR2 at HSCT, Gilleece et al. similarly found higher 2-year relapse rates in MRD-positive (40%) versus MRD-negative (24%) patients [65]; however, no significant difference was found in overall survival among the two groups. Finally, in a retrospective study of 580 patients with AML who underwent allo-HSCT in CR1 or CR2, MRD status at HSCT remained an independent poor prognostic marker regardless of remission number [66]. In these studies, MRD status has been repeatedly identified as a key prognostic marker of allo-HSCT outcomes at both CR1 and CR2.

The effect of MRD status prior to HSCT may also be dependent on ELN risk category. For example, in a retrospective cohort of 176 patients with AML, Jentzsch et al. identified MRD-positive status prior to HSCT as a significant factor for relapse in the ELN favorable and intermediate groups, but not in the adverse group [67]. They further demonstrate a graded benefit (specifically, an increase in time from allo-HSCT to relapse) by ELN risk group for MRD-negative patients compared to their MRD-positive counterparts: 3.6 years in favorable, 2.1 years in intermediate, and 0.5 years in adverse risk groups [67]. Overall, though these studies reflect a benefit to MRD eradication prior to allo-HSCT, they also show that some patients can achieve long-term survival even when transplanted in suboptimal (i.e., MRD-positive) remission states (Figure 2, scenario D).

### 5.2. For Whom Is Transplant Indicated?

Beyond its use as a prognostic marker, the application of MRD status in transplant decisions has raised several questions. Should only MRD-positive patients undergo transplant since they have the most to gain? Conversely, should only MRD-negative patients undergo transplant given they are more likely to remain remission-free? Should we delay transplant in an effort to achieve MRD eradication? Figure 2 scenarios A–D highlight ongoing debate around these questions.

The application of MRD in transplant may be the most useful in patients with intermediate risk AML. For example, the GIMEMA AML1310 randomized control trial compared outcomes in favorable and intermediate risk patients [21]. Among the MRD-positive intermediate risk patients who underwent allo-HSCT, outcomes were similar to favorable risk patients (2-year overall survival was 58% and disease-free survival was 61%), suggesting a strong benefit to transplant. In a retrospective analysis of AML17 trial data, patients with AML without the *NPM1* mutation demonstrated more benefit from allo-HSCT if MRD-positive (HR 0.72) than MRD-negative (HR 1.68) [68]. For patients with non-favorable risk AML and *NPM1* mutation, the ALFA-0702 trial found that allo-HSCT improved disease-free survival (HR 0.25) and overall survival (HR 0.25) in patients with less than 4-log reduction in *NPM1* MRD after initial chemotherapy [69]. These studies pose the questions: Can we improve outcomes by treating patients with MRD-positive intermediate risk AML as adverse risk? Should we treat MRD-negative intermediate risk AML as favorable risk?

Few studies explore transplant benefit in patients with persistent MRD and favorable risk AML. The prospective AML05 trial reported allo-HSCT outcomes in patients with favorable risk t(8;21) AML based on MRD response after the second consolidation [70]. In patients with persistent MRD (<0.3-log reduction in *RUNX1::RUNX1T1* transcripts), allo-HSCT resulted in a lower cumulative incidence of relapse (22.1% vs. 78.9%) and improved disease-free survival (61.7% vs. 19.6%) compared to chemotherapy alone at 4 months after CR. Taken together, these studies show that allo-HSCT may have more benefit in patients with MRD-positive rather than MRD-negative intermediate and possibly even MRD-positive favorable risk AML. More randomized control trials are needed [71,72].

### 5.3. Can We Use MRD to Determine Optimal Conditioning Regimen?

The impact of conditioning regimen prior to allo-HSCT in MRD-positive patients remains controversial. In an early retrospective study, Walter et al. showed no significant difference in MRD-positive recipients of a myeloablative vs. non-myeloablative regimen in terms of 3-year relapse (63% vs. 57%) and 3-year overall survival (25% vs. 41%) [73]. In a subsequent larger study, the same group demonstrated no impact of the conditioning regimen (including myeloablative, reduced intensity, and non-myeloablative) on 3-year relapse (69% vs. 57% vs. 57%), relapse-free survival (18% vs. 24% vs. 20%), or overall survival (33% vs. 30% vs. 31%) in MRD-positive patients [74]. They also identified no effect of total body irradiation in myeloablative conditioning in AML based on MRD status [75].

In contrast, in a randomized control trial, Hourigan et al. showed that MRD-positive patients had lower 3-year cumulative relapse (19% vs. 67%) and improved 3-year overall survival (61% vs. 43%) in recipients of a myeloablative conditioning regimen as opposed to reduced-intensity conditioning prior to allo-HSCT [28]. Based on these results, Freeman et al. postulate that pretransplant MRD-negative patients may be saved from the extra toxicity of the myeloablative regimen or even allo-HSCT itself (i.e., treated as favorable risk patients) [76]. As many MRD-positive patients will not be eligible for myeloablative conditioning, Freeman et al. also advocate for novel strategies to reduce the risk of an early disease relapse after low intensity conditioning [76].

### 5.4. Can We Use MRD to Determine Optimal Donor Type?

Few prospective studies compare outcomes after haploidentical (haplo-HSCT) versus HLA-matched sibling donor transplant (MSDT) [77,78]. Among patients who were MRD-positive prior to transplant in a study that combined retrospective and prospective data, haplo-HSCT was associated with lower relapse (19% vs. 55%) and improved overall survival (83% vs. 38%) at 4 years when compared to MSDT [79]. A subsequent prospective study compared the two transplant types with the primary end point of post-transplant MRD. Though a comparable number of patients were MRD-positive prior to transplant (33% vs. 29% in haplo-HSCT% vs. MSDT cohorts), patients who received haplo-HSCT had lower rates of post-transplant MRD compared to MSDT (18% vs. 42%) [80]. This difference was attributed to a greater graft-versus-leukemia effect in haplo-HSCT.

### 5.5. How Should We Treat MRD Relapse during Post-Transplant Surveillance and/or Chronic Low Intensity Therapy?

In patients with adverse risk AML who have undergone HSCT, survival rates remain low at around 55%, and disease relapse remains the leading cause of death [81]. Modalities to address MRD relapse and augment graft versus leukemia effect after transplant include MRD-directed treatments (i.e., hypomethylating agents, venetoclax, and inhibitor therapies), reduction of immunosuppression, donor lymphocyte infusions (DLI), salvage chemotherapy, and second allo-HSCT. Figure 2 scenario E highlights the undefined approach to this clinical scenario.

The hypomethylating agents azacitidine and decitabine have been studied as preemptive therapies in patients with AML after allo-HSCT. For example, in the prospective RELAZA trial, MRD-triggered treatment with azacitidine was found to be an effective strategy in preventing or substantially delaying hematologic relapse in patients with MRD after allo-HSCT [82]. In another randomized multicenter trial, treatment with low dose decitabine with recombinant human G-CSF resulted in a lower 2-year cumulative incidence of relapse compared to no treatment (15.0% vs. 38.3%) in MRD-negative patients with other high-risk features of AML after allo-HSCT [81]. As shown in Figure 2, scenario F, whether and when to discontinue these treatments remains unknown.

For patients with known molecular mutations, inhibitor therapies may have a benefit for MRD prevention and treatment. For example, for patients with *FLT3-ITD*-mutated AML, the *FLT3* inhibitors gilteritinib, quizartinib, and sorafenib have all been found to improve patient outcomes as maintenance therapy after allo-HSCT in the r/r setting [83,84,85,86,87,88]. For those who achieve undetectable *FLT3*, the European Society for Blood and Marrow Transplantation suggests cessation of *FTL3* inhibitors at 2 years [88].

Finally, treatment for patients with MRD relapse after allo-HSCT can include matched DLI with the reduction or withdrawal of immunosuppression, as reviewed elsewhere [89]. Notably, the Acute Leukemia Working Party of the European Society for Blood and Marrow Transplantation recommends the use of pre-emptive matched DLI for patients with evidence of MRD post-allo-HSCT [90]. This practice is also supported by expert opinion [89,91].

## 6. Conclusions

In this commentary, we discuss the role of MRD in AML, propose a framework for MRD monitoring, and identify outstanding questions regarding the application of MRD in AML treatment. MRD remains a significant marker of poor outcomes in AML. There are, however, several promising approaches to preemptive and targeted therapies directed to MRD, such as hypomethylating agents, venetoclax, and inhibitor therapies. Though a majority of studies we address dichotomize MRD status, the increasing use of high-precision molecular techniques suggest that MRD can be better understood along a spectrum of depth (i.e., complete molecular remission versus molecular persistence at a certain copy number) [4].

Available data on the role of MRD in AML emphasize the need for prospective randomized trials to address ongoing questions. Is there a treatment, such as hypomethylating agents, that can serve as an MRD “eraser” in AML, and would that matter for outcomes? How should we change management in the setting of MRD persistence, progression, or relapse, either before or after transplant? Can MRD-negative status justify the avoidance of allo-HSCT in some patients, or adjustment/discontinuation of chronic low-intensity therapy for others?

Studies addressing these questions should be designed such that MRD is a treatment endpoint, given its evident prognostic significance, role in treatment selection, and potential to be a surrogate of overall survival [50,92]. For example, patients with active AML could be combined with patients in MRD-positive remission, given they have a similarly high-risk disease [17,93,94]. Meanwhile, patients in MRD-negative remission are ideal for prospective, controlled studies evaluating which therapies improve outcomes in lower risk patients [95]. In these studies, we must continue to consider patient and system barriers, including limited finances, healthcare literacy, social support, donor availability, comorbidity burden, and transplant center access [96].

## Figures and Tables

**Figure 1 cancers-14-03634-f001:**
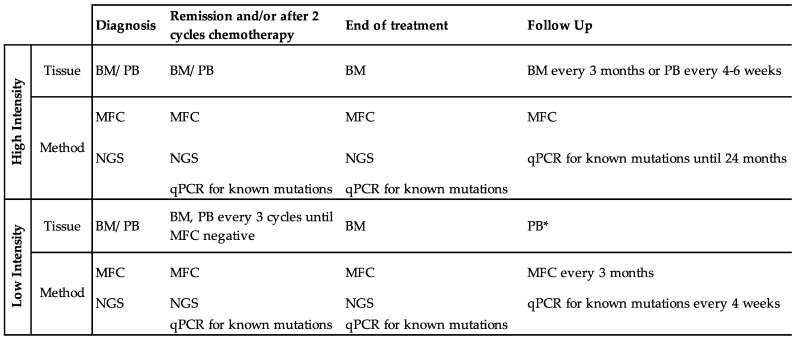
MRD Monitoring in AML. Recommendations adapted from the 2021 European LeukemiaNet Guidelines [10], the National Comprehensive Cancer Network AML Guidelines version 1.2022 [11], and our clinical experience and emerging data on lower intensity regimens [38]. Tissues for MRD testing include peripheral blood (PB) and bone marrow (BM). MRD quantification techniques include multiparameter flow cytometry (MFC), next-generation sequencing (NGS), and quantitative polymerase chain reaction (qPCR). * Bone marrow studies can be obtained as clinically indicated in patients on continuation/maintenance therapy or after treatment completion.

**Figure 2 cancers-14-03634-f002:**
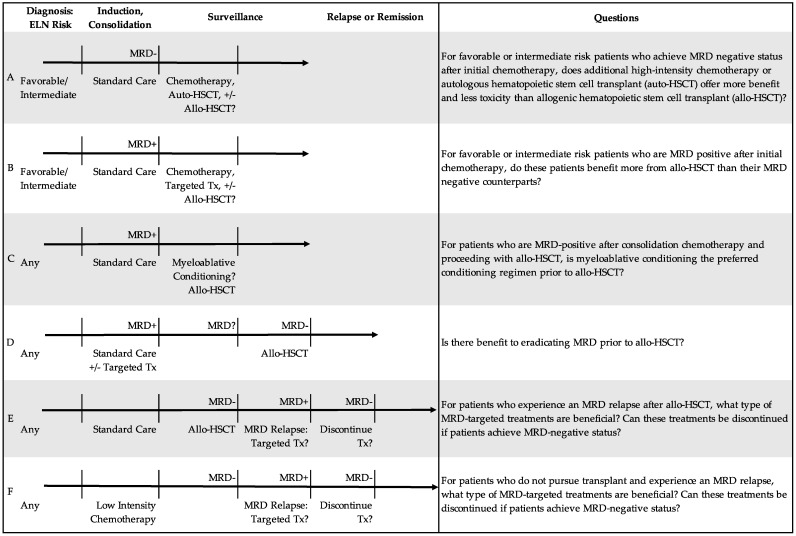
Six scenarios of AML treatment based on European LeukemiaNet (ELN) risk category and measurable residual disease (MRD) status. ELN risk categories are noted in the first column [3]. MRD status is first established after achieving remission and up to two cycles of chemotherapy, as defined in the ELN consensus 2021 update [10]. As the most significant timepoint of MRD-negative state in relation to treatment administered remains unknown, we show MRD status after initial induction and/or consolidation chemotherapy. Abbreviations: measurable residual disease positive (MRD+); measurable residual disease negative (MRD-); treatment (tx); allogeneic hematopoietic stem cell transplant (allo-HSCT); autologous hematopoietic stem cell transplant (auto-HSCT).

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
