# Peer review of "Clinical Impact of Measurable Residual Disease in Acute Myeloid Leukemia"

_cancers, 2022, doi:10.3390/cancers14153634_

Round 1

Reviewer 1 Report

my comments have been sufficiently addressed.

please italicise "WT1" (line 96). 

Reviewer 2 Report

I have no further comments, this revised version of the paper is suitable for pubblication.

Reviewer 3 Report

This is an important topic and the manuscript/ content has improved from the last version.

Thank You 

This manuscript is a resubmission of an earlier submission. The following is a list of the peer review reports and author responses from that submission.

Round 1

Reviewer 1 Report

Thank you for the opportunity to review the submitted manuscript “Clinical Impact of Measurable residual disease in Acute Myeloid Leukemia” submitted by Azenkot and Jonas to Cancers. Although I think that MRD in AML remains a clinically highly relevant topic, I think it is not suitable for publication in its current form. Reasons mainly include a lack of accuracy, however, I would also wish for a broader discussion of different findings rather than “just” citing one paper after the other.

Please find attached my comments:

Major points:

·      please italicize all human genes 

·      please use the correct gene name (BCR::ABL1 instead of BCR-ABL)

·      would suggest to adhere to the recently published HUGO Gene Nomenclature Committee recommendations that advocates a double colon as unique separator to be used in the description of fusion genes (e.g. PML::RARA)

·      methods: “by selecting high impact journals including…” while it is understandable that high impact journals are more frequently read and cited, a) I would expect a more comprehensive approach from a review also including other relevant journals including but not limited to Hemasphere (official EHA journal), Blood Advances etc, as the reader should receive a broad overview on the topic rather than a selection of a few highest-impact papers, b) the current impact factors of the American Journal of Hematology (10.047) is higher than that of Haematologica (9.941), and c) the paper is submitted to cancers (IF 6.6!)

·      3.1.

o   I clearly miss the discussion on which markers should be tested: while MFC can be applied in the majority of patients with AML, it clearly has a lower sensitivity (also frequently lower than the mentioned 10-4) than certain well-established qPCR methods. The ELN guidelines clearly state that molecular markers (NPM1, RUNX1::RUNX1T1, CBFB::MYH11) are to be preferred over MFC when detected in a patient and I believe that this is the clinical standard in most hospitals. 

o   “qPCR assays have been developed for three phenotypes […] NPM1, CBF, PML-RARA” ïƒ  the authors refer to genetic aberrations in AML (which are no phenotypes). “CBF” are the core binding factor leukemias, which per definition constitute two distinct gene fusions (RUNX1::RUNX1T1, CBFB::MYH11) which result in distinct MRD assays. This should be precisely and correctly discussed in a review on MRD.

o   Targets for PCR assays: here, gene fusions and WT1 (where in this context I believe the gene expression, rather than the gene mutation is meant) are mentioned together, but also these assays result in very different sensitivities: gene fusions are specific for leukemia, while WT1 is a gene expressed also in healthy individuals. While I agree that WT1 represents a MRD marker when overexpressed (together with other AML-associated gene expressions, as BAALC or MN1), these limitations should be discussed and not just mentioned together with other markers (which are fusion genes)

o   Costs: directly after addressing costs in healthcare, the authors mention changing cytogenetics to WES (which would be not at all cost-effective)

·      3.2.

o   “known mutations in NPM1, CBF and APL” ïƒ  this is only correct for NPM1. CBF is NOT a gene and NOT one entity, it refers to different possible gene fusions in AML. Similarly, APL is a disease and its underlying genetic aberration is a fusion gene and not a mutation

o   “the role of MRD in patients with relapsed/refractory AML is less clear.” And “it remains unknown if MRD-negative status prior to allo-HSCT impacts outcomes in either the frontline or r/r setting” ïƒ a lot of studies analyzed the impact of MRD at allo HSCT, including comparison of MRD to morphologic remission (e.g. Araki et al, JCO; Jentzsch et al, Blood Cancer J) and including in first or second remission (e.g. Walter et al, Blood 2013; Gilleece et al, Blood Cancer J 2021; Jentzsch et al. Blood Adv 2022). These can be discussed, rather than merely stating that it remains unclear. In contrast, a patient with r/r AML (= active disease) does not need a MRD analysis as he has active disease.

·      Table 1: 

o   there is a known clear difference in MRD assessment in PB/KM, most studies suggest appr. 1 log lower sensitivity for PB. Additionally, MRD every 4 weeks will clearly not address a BM analysis. This should be included/discussed, e.g. that a higher sampling frequency may mitigate the lower sensitivity in PB compared to less often BM.

o   What does * refer to?

·      3.3

o   Non-DTA CHIP mutations may be MRD markers: please be more clear which mutations are meant and how the data contributes to this statement. there are studies evaluating other CHIP mutations showing high persistence rates in remission and no outcome impact for e.g. SRSF2(Grimm et al, Am J Hematol 2021) or IDH2 R140Q (Bill et al, Blood Advances 2022)

·      4.3

o   I miss a discussion considering the lead time bias in MRD-positive patients receiving preemptive treatment

o   As mentioned above, a variety of studies showed that MRD at allo HSCT is highly predictive for outcomes. The statement “it is unknown whether MRD eradication is indicated prior to allo Tx” should be discussed in more-depth. 

·      5.5

o   Relaza is no maintenance, but a preemptive therapy concept

Minor points

·      Please consider using “higher risk” instead of “greater risk” 

·      3.1 “in their recent review, Short et al durther describe MRD detection methods”… I do not find this statement very useful when reading this review.

·      3.1 NGS panels: specify RAS (NRAS, KRAS, HRAS?)

·      3.2 “the optimal timing of MRD conversion […] remains unknown ïƒ  please specify. Clearly we believe the earlier a patients gets negative the better…?

·      after abbreviating allo HSCT, do not use “allogeneic stem cell transplant” afterwards

·      abbreviate DLI the first time used in the text

·      correct typos … e.g. Figure 2 “relpase” or “indeterminant” in 3.3

Reviewer 2 Report

In this paper, the Authors provide a commentary on the MRD assessment acute myeloid leukemia, providing an up-to date review. Different methods of MRD assessment are considered, analyzing theri sensitivity and the most adeguate timing of assessment. The paper also examinates the role of MRD assessment in initial therapy, follow up and on stem cell transplantation and in the last section several open questions are addressed.

Overall, the paper is clearly written and coincise. I think that it is useful as it provides a comprehensive and rapid view of the current landscape of MRD assessment.

I think that no major changes are required.

Reviewer 3 Report

In the commentary: ”Clinical Impact of Measurable Residual Disease in Acute Myeloid Leukemia” by Azenkot and Jonas the authors are summarizing some aspects of MRD in AML. This is a topic of high clinical impact and interest. 

However, my first question would be, why is this manuscript a “commentary” and not a “review”? If it is a “commentary” which in my opinion may be more subjective and may also include personal opinions, it should be made clearer and also include such statements. If it is a review it should be more comprehensive.

The authors claim that the review was based on a pubmed search for high impact oncology journals -what are these journals and what is the impact threshold – were only Oncology subspeciality journals? NEJM would not be one of those. It remains quite unclear what has been included and why or why not, e.g. why not the American journal of hematology? Or e.g. BMT for transplant setting? Even if this is further defined one might doubt that this is a good way to sort out papers for a review. A broader approach may be more helpful given the fact that in the past some abstract or papers first published in less well established journals turned out to be highly impactful – these may be missed. In my opinion the manuscript may be improved by a more comprehensive view of this complex topic. Matters of sensitivity regarding BM vs PB and gDNA vs cDNA should be discussed. The depth of MRD also regarding the different techniques should be discussed, since this may have great impact on study results and clinical discissions (e.g. NGS vs Flow vs qPCR/dPCR). Some additional effort in summarizing the “knowns” und “unkowns” would be appreciated. Some words towards different aberrations and the MRD meanings (not just DTA vs non-DTA in Tx vs no Tx) and also e.g. relapse dynamics etc would be worth discussing.

For Figure 1 what is the basis of the recommendations. What is a low or a high intensity treatment – is the aggressiveness of the observed AML phenotype not of greater importance than the treatment aggressiveness (especially in time with increasing targeted therapies)

Please italicize genes. Please use correct spelling (HUGO) of genes (e.g. ASXL1 and ABL1). Please use “::” instead of “-“ in fusions (e.g. BCR::ABL1)

In the simple summary I am not sure what “disease severity” is and how it relates to the MRD amount. Maybe something like: remaining burden / amount would better describe it.

A note for the first sentence in 4.1.: “MRD is a poor prognostic factor in AML” is misleading and maybe misunderstood - since it actually is a quite good prognostic marker for worse outcome. 
